# Flow Cytometric Assessment of FcγRIIIa-V158F Polymorphisms and NK Cell Mediated ADCC Revealed Reduced NK Cell Functionality in Colorectal Cancer Patients

**DOI:** 10.3390/cells14010032

**Published:** 2024-12-31

**Authors:** Phillip Schiele, Stefan Kolling, Stanislav Rosnev, Charlotte Junkuhn, Anna Luzie Walter, Jobst Christian von Einem, Sebastian Stintzing, Wenzel Schöning, Igor Maximilian Sauer, Dominik Paul Modest, Kathrin Heinrich, Lena Weiss, Volker Heinemann, Lars Bullinger, Marco Frentsch, Il-Kang Na

**Affiliations:** 1BIH Center for Regenerative Therapies (BCRT), Therapy-Induced Remodeling in Immuno-Oncology, Berlin Institute of Health at Charité—Universitätsmedizin Berlin, 13353 Berlin, Germany; 2Department of Hematology, Oncology and Cancer Immunology, Charité—Universitätsmedizin Berlin, Corporate Member of Freie Universität Berlin and Humboldt—Universität zu Berlin, 13353 Berlin, Germany; 3BIH Biomedical Innovation Academy, BIH Charité Junior Digital Clinician Scientist Program, Berlin Institute of Health at Charité—Universitätsmedizin Berlin, 10178 Berlin, Germany; 4BSIO Berlin School of Integrative Oncology, Charité—Universitätsmedizin Berlin, Corporate Member of Freie Universität Berlin and Humboldt-Universität zu Berlin, 10178 Berlin, Germany; 5Medical Department of Hematology, Oncology and Tumor Immunology, Molekulares Krebsforschungszentrum (MKFZ), Charité—Universitätsmedizin, 10117 Berlin, Germany; 6MVZ Onkologie Tiergarten, 10559 Berlin, Germany; 7Department of Surgery, Campus Charité Mitte—Universitätsmedizin Berlin, Corporate Member of Freie Universität Berlin and Humboldt—Universität zu Berlin, 10117 Berlin, Germany; 8Department of Surgery, Campus Virchow Klinikum, Charité—Universitätsmedizin Berlin, Corporate Member of Freie Universität Berlin and Humboldt—Universität zu Berlin, 13353 Berlin, Germany; 9German Cancer Consortium (DKTK), 10115 Berlin, Germany; 10Department of Medicine III, Ludwig-Maximilians-University of Munich, 80539 Munich, Germany; 11Department of Hematology/Oncology and Comprehensive Cancer Center, University Hospital, Klinikum Grosshadern, Ludwig-Maximilians-University of Munich, 80539 Munich, Germany; 12ECRC Experimental and Clinical Research Center, Charité—Universitätsmedizin Berlin, Corporate Member of Freie Universität Berlin and Humboldt—Universität zu Berlin, 10178 Berlin, Germany; 13Max Delbrück Center for Molecular Medicine in the Helmholtz Association (MDC), 13125 Berlin, Germany

**Keywords:** antibody-dependent cell cytotoxicity, cetuximab, colorectal neoplasms, flow cytometry, single nucleotide polymorphism, natural killer cells

## Abstract

Antibody-dependent cell-mediated cytotoxicity (ADCC) by NK cells is a key mechanism in anti-cancer therapies with monoclonal antibodies, including cetuximab (EGFR-targeting) and avelumab (PDL1-targeting). Fc gamma receptor IIIa (FcγRIIIa) polymorphisms impact ADCC, yet their clinical relevance in NK cell functionality remains debated. We developed two complementary flow cytometry assays: one to predict the FcγRIIIa-V158F polymorphism using a machine learning model, and a 15-color flow cytometry panel to assess antibody-induced NK cell functionality and cancer-immune cell interactions. Samples were collected from healthy donors and metastatic colorectal cancer (mCRC) patients from the FIRE-6-Avelumab phase II study. The machine learning model accurately predicted the FcγRIIIa-V158F polymorphism in 94% of samples. FF homozygous patients showed diminished cetuximab-mediated ADCC compared to VF or VV carriers. In mCRC patients, NK cell dysfunctions were evident as impaired ADCC, decreased CD16 downregulation, and reduced CD137/CD107a induction. Elevated PD1+ NK cell levels, reduced lysis of PDL1-expressing CRC cells and improved NK cell activation in combination with the PDL1-targeting avelumab indicate that the PD1-PDL1 axis contributes to impaired cetuximab-induced NK cell function. Together, these optimized assays effectively identify NK cell dysfunctions in mCRC patients and offer potential for broader application in evaluating NK cell functionality across cancers and therapeutic settings.

## 1. Introduction

The introduction of monoclonal antibodies (mAbs) dramatically expanded therapy options for cancer patients. Besides blockade of specific signaling pathways defined by their antigen binding site, antibodies of the immunoglobulin G1 (IgG1) isotype can engage with Fc-gamma receptors (FcγRs), especially the FcγRIIIa (also known as CD16), expressed on immune cells such as natural killer (NK) cells to induce antibody-dependent cell-mediated cytotoxicity (ADCC) in targeted cells [1].

NK cell responses mediated by the FcγRIIIa are of major importance for the efficacy of therapeutic monoclonal antibodies [2]. A single nucleotide polymorphism (SNP) in the *FCGR3A* gene results in FcγRIIIa variants with either a valine (V) or phenylalanine (F) residue at amino acid position 158 (FcγRIIIa-V158F), also known as FcγRIIIa-158V/F polymorphism [3]. Since IgG1 binds in proximity to amino acid 158, the polymorphism affects IgG1-binding by FcγRIIIa. Compared to F-F homozygotes, FcγRIIIa-158-V-V subjects possess NK cells with a higher affinity to IgG1 and, in turn, the ability to mediate a stronger ADCC [3]. Consequently, in hematologic cancers treated with the anti-CD20 mAb rituximab, the presence of the high-affinity FcγRIIIa-158-V-V polymorphism is associated with improved treatment response [4]. These findings already led to the approval of an anti-CD20 mAb with enhanced capability to bind FcγRIIIa, especially to the low-affinity F-F variant, to cover a broader patient population [5]. However, the influence of FcγRIIIa polymorphisms seems to be more ambiguous in solid cancers. In metastatic colorectal cancer (mCRC) several studies have considered the FcγRIIIa-V158F polymorphism as a relevant factor for efficacy and prediction in clinical trials. Currently, chemotherapies containing the epidermal growth factor receptor (EGFR) targeting mAb cetuximab are the standard therapy in patients with *RAS* wild-type, left-sided mCRC [6] which yielded both negative and positive associations between FcγRIIIa phenotypes and progression-free or overall survival [7,8,9,10]. While positive clinical correlations with the high-affinity VV phenotype would be intuitive, several studies indicate that the selective immunologic pressure on cancer cells [9], activation of tumor associated-macrophages [11] or the preferred engagement with the inhibitory FcγRIIb [7] could also translate into opposite effects. Hence, its impact is still highly controversial and needs further investigation.

By binding to the EGFR expressed in cancer cells, cetuximab blocks the ligand binding site, preventing receptor dimerization and downstream activation. However, comparisons of clinical studies in head and neck cancers suggest that immune-mediated effects might be the main driver for effective anti-tumor responses of cetuximab [12]. Through its IgG1 isotype, the induction of ADCC by cetuximab is well-documented in vitro [13] and it was also shown to increase the cytotoxicity of immune cells during the treatment of colorectal cancer patients [14]. In addition to direct anti-cancer effects through ADCC, engagement of cetuximab with FcγRs was shown to improve immune crosstalk through the release of antigens by induction of immunogenic cell death or maturation and activation of antigen-presenting cells resulting in effective anti-tumor responses by innate and adoptive immune cells [15]. Since increased immune activation was shown to also induce immunosuppressive feedback mechanisms, e.g., the upregulation of the immune checkpoint ligand PDL1 [16], several clinical studies such as the FIRE-6 trial [17] currently investigate potential synergies between cetuximab and immune checkpoint blockade.

Here, we report the development of a machine learning model to determine the FcyRIIIa-V158F polymorphism based on flow cytometric measurements. We subsequently established a co-culture assay and a 15-color flow cytometry panel for the evaluation of antibody-induced immune responses during cancer therapy. Our set-up was established on peripheral blood mononuclear cells (PBMC) from healthy donors as well as from mCRC patients before and during antibody-based treatment and was able to identify correlations between surface marker expressions and ADCC sensitivity. Finally, we could show that immune cells from mCRC patients present an impaired reactivity against CRC cells compared to matched healthy controls.

## 2. Materials and Methods

### 2.1. Origin of PBMCs from mCRC Patients and Healthy Volunteers

Clinical samples were obtained from mCRC patients recruited within the non-randomized, single-arm, multi-center, phase-II FIRE6 study (EudraCT 2018-002010-12). A total of 52 patients with previously untreated RAS/BRAF wild-type mCRC were enrolled and samples used in the presented study were obtained prior to therapy initiation (baseline, C1), after one (C2) or four (C5) cycles of FOLFIRI+ cetuximab or after one additional cycle of combination therapy also containing avelumab (C6). To establish the flow cytometry panel to predict FcγRIIIa polymorphisms, we collected samples from 29 randomly chosen healthy donors. In addition, 10 healthy volunteers, matched to the age and FcγRIIIa distribution of patients from the FIRE-6 study were recruited. Symptoms of acute infection at the time of blood withdrawal were ruled out for all volunteers by anamnesis. Clinical characteristics of mCRC patients and healthy donors are summarized in Appendix A. Acquisition of blood samples and PBMCs from patients and healthy volunteers was approved by the ethics committees of the Ludwig-Maximilians-University of Munich and the Charité—Universitätsmedizin Berlin.

### 2.2. Isolation and Handling of Human PBMCs

PBMCs were isolated from heparinized whole blood by density centrifugation within 2 days after blood withdrawal. To do so, blood was mixed 1:1 with phosphate buffered saline (PBS), pipetted onto BioColl separation solution (Bio&SELL, Nürnberg, Germany, #L6115), and centrifuged for 20 min at 800× *g*. After washing, cells were counted on a LUNA-FL™ Dual Fluorescence Cell Counter (BioCat, Heidelberg Germany) and subsequently resuspended in freezing medium (15% RPMI1640 medium (Gibco, New York, NY, USA, #12055), 75% fetal bovine serum (FBS) (Sigma, Livonia, MI, USA #F7524), and 10% dimethyl sulfoxide (DMSO) (Sigma Aldrich, St. Louis, MO, USA #D2650)). Cell were aliquoted at 2.5 × 10^6^ cells/vial and, after rate-controlled cooling to −80 °C for 24 h, transferred to cryostorage until use. For controlled thawing, cryopreserved PBMCs were quickly thawed by placing them for 2 min in a 37 °C water bath and subsequently transferred drop-wise into 10 mL pre-warmed RPMI1640 containing 20% FBS. After two additional washing steps in media, PBMCs were counted and adjusted to conduct the respective assays.

### 2.3. Flow Cytometric Analyses

Throughout this work, the flow cytometric measurements of all experimental samples were exclusively acquired using a CytoFLEX LX cytometer (Beckman Coulter, Miami, FL, USA). Instrument performance was monitored daily with CytoFLEX daily QC fluorospheres (Beckman Coulter, #B53230). A defined volume of 150 µL per sample was measured at a constant acquisition rate of 150 µL/min, keeping the abort rate < 1% over the course of the measurement. Prior to use, every antibody of the flow cytometry panels was titrated to verify its reactivity and to determine the optimal signal/noise ratio (Appendix A). Antibody information for the flow cytometry panels to assess FcγRIIIa polymorphisms, cell line characterization, or ex vivo ADCC activity is depicted in Appendix A, respectively. Flow cytometry data were analyzed using CytExpert Software Version 2.4 (Beckman Coulter, Brea, CA, USA) and FlowJo Version 10.8.1 (Becton Dickinson, Franklin Lakes, NJ, USA).

### 2.4. FcγRIIIa Detection

To establish and validate the detection of FcγRIIIa polymorphisms by flow cytometry we first genotyped all tested samples by assessing the G559T single nucleotide polymorphism (SNP). DNA from 10^5^ PBMCs was isolated using the QIAamp UCP DNA Micro Kit (Qiagen, Hilden, Germany #56204) on a QIAcube (Qiagen, #QIAG990395). PCR amplification and melting curve analysis using the High-Resolution Melting Kit (Roche Life Science, Rotkreuz, Switzerland #50-720-3243) on a Light-Cycler 480 instrument (Roche Life Science) were carried out externally as a paid service by the FyoniBio GmbH (Berlin, Germany) to form the ground truth for our analyses.

For flow cytometric analyses of the FcγRIIIa-158V/F polymorphisms, 10^5^ PBMCs were stained for 15 min with CD3, CD14, CD56 and two different antibody clones against CD16. After doublet and debris exclusion, we exploit the binding epitopes of two different CD16 antibody clones (MEM and LNK, see Appendix A) to detect FcγRIIIa-158V/F polymorphisms on CD3-CD14-CD56+ NK cells (Figure 1B).

From the flow cytometry data of 91 individuals (39 healthy donors and 52 FIRE6 patients), we extracted 30 flow cytometry parameters to develop a supervised prediction model based on the FcγRIIIa genotypes as determined by the PCR method (feature construction step). Most of those parameters belong to the binding strength (frequencies, median fluorescence intensity (MFI)) of the two CD16 clones on NK cell subsets (Appendix A). We applied a conditional random forest model to select the five most important parameters to distinguish between FcγRIIIa genotypes (feature selection step). To reduce the problems due to multicollinearity of parameters introduced by the combination of multiple flow cytometry measures, we calculated the variance inflation factor (VIF) for each of the five features and set an upper threshold of VIF < 5 (multicollinearity reduction step). After this pre-processing, the MFI ratio of MEM154 to LNK16 on CD56dimCD16+ NK cells (rMFI_NKdimML_) and the frequency ratio of CD56dimCD16+ NK cells (as a percentage of all NK cells) between MEM154 and LNK16 (rF_NKdimML_) were identified as the most distinctive parameters. Next, these parameters were fed into a supervised prediction model consisting of a Linear Discriminant Analysis (LDA) to maximize between-class differences within FcγRIIIa genotypes followed by Logistic Regression. The mean prediction skill of the model as reflected by the F1 scores was evaluated by 10-fold cross-validation using 91 samples, consisting of 39 healthy donors as well as samples from 52 FIRE6 patients acquired before the start of treatment (time point C1). For 10-fold cross-validation, the dataset is split into 10 equal groups and at each iteration one group is used once as a test set while the remaining nine groups are taken as the training set. In addition, samples of the FIRE6 patients at 3 further time points during treatment (time points C2, C5 and C6) were acquired and F1 scores and weighted F1 scores, integrating class imbalances, for each time point were evaluated by a modified leave-one-out cross-validation (LOOCV). In LOOCV, a single sample is used as test set at each iteration and all other samples make up the training set. In our case the procedure had to be modified to exclude the samples at all other time points of the selected test patient for a given iteration in order to prevent data leakage from the test to the training set.

We established the following acceptance criteria to compare the G559T genotype PCR results with the flow cytometry-based FcγRIIIa polymorphism data: (1) weighted F1 scores of the flow-cytometry-based-approach should be ≥90% across all samples and (2) no misclassifications must occur between the low-affinity FF and the high-affinity VF/VV variants as this distinction appears to be most clinically relevant.

### 2.5. Cancer Cell Lines

For this study, 13 different CRC cell lines (Appendix A) were cultured in DMEM (Gibco, #21885-025), RPMI-1640 (Gibco, #310870-025) or Leibovitz’s L-15 medium (Sigma, #L5520), supplemented with 10% FBS (Sigma, #F7524), 1% Penicillin-Streptomycin (P/S; Sigma, #P4333) and 1% GlutaMax^TM^ (Gibco, #35050061). For characterization, each cell line was screened for the expression of nine different surface markers covering EpCAM for identification of cancer cells in the ADCC assay, targets for therapeutic antibodies (EGFR, PDL1) and markers known to be involved in NK cell functionality and ADCC when expressed on targeted cells (HLA-A,B,C, HLA-E, MICA/B, CD137L, CD40, OX40L; Appendix A). Cells were stained in single-staining for 15 min with fluorophore-conjugated antibodies against each marker or a respective isotype control antibody to normalize the frequency of positive cells or the MFI. All tested cell lines were routinely tested negative for mycoplasma contamination (MycoAlertTM Mycoplasma Detection Kit, Lonza, Basel, Switzerland #LT07-318) and cultured for a maximum of 30 passages before renewal.

### 2.6. ADCC Assay

Assessment of ADCC and additional antibody-induced anti-cancer immune response was analyzed after 24 h co-cultures of PBMCs from healthy donors or mCRC patients with CRC cell lines by LDH release and flow cytometry. After titration experiments of cancer cell counts (Appendix A) to identify a range concerning a good assay sensitivity while limiting the number of effector cells needed, we selected 3 × 10^4^ cancer cells as the lowest amount where changes in cell viability are reliably detectable by LDH release and flow cytometry. Comparisons of different effector:target (E:T) ratios (Appendix A) and concentrations of cetuximab and avelumab (Appendix A) revealed our final assay conditions using a 10:1 E:T ratio and 100 ng/mL of the tested antibodies. Co-cultures were performed in 96-well flat-bottom plates using phenol-red free DMEM (Gibco, #11880028) with only 1% fetal bovine serum to allow for LDH downstream analysis and 1% Triton-X-100 was added as positive control. Additionally, the CD107a antibody was directly added at the start of co-culturing while the protein-transport inhibitor GolgiStop (BD BioSciences, Franklin Lakes, NJ, USA #554724) was added in 1:2500 dilution 4 h before the end of incubation in order to detect NK cell degranulation. To reduce potential spill-over or evaporation during cultivation, wells were sealed with an adhesive foil (Thermo Scientific, Waltham, MA, USA #10130853) and placed in an incubator at 37 °C, 5% CO_2_, and 95% relative humidity for 24 h. In general, experiments were performed in technical triplicates but had to be adjusted in some cases of scarce patient material. In each experiment, we included triplicates of only medium in order to correct for the background signal deriving from the medium. LDH release was determined using the commercially available Cytotoxicity Detection Kit (Roche, Basel, Switzerland #11644793001), according to the manufacturer’s protocol after 25 min incubation of 75 µL supernatant with the LDH mix. Absorbance was recorded using a microplate spectrophotometer system (Tecan) and specific lysis was calculated as follows: specific lysis[%] = ([PBMCs + cancer cells + antibody] − [PBMCs + cancer cells])/([cancer cells + Triton] − [cancer cells alone]) × 100%.

Since all tested CRC cell lines grow adherently, they need to be detached prior to flow cytometry. Therefore, the remaining supernatant including PBMCs was transferred to a 96-well round bottom plate and cancer cells were gently detached with 50 µL TrypLE (Life Technologies, Waltham, MA, USA #12604-013) for 15 min. Detached cells were transferred to the respective wells of a 96-well round-bottom plate and centrifuged at 300× *g* for 5 min. Following centrifugation, cells were stained in 50 µL of Annexin V Binding Buffer (BioLegend, San Diego, CA, USA #422201) with a panel of 15 fluorochrome-conjugated antibodies (Appendix A) for 15 min. Subsequently, cells were washed once, resuspended in 150 µL Annexin V Binding Buffer and a defined volume of 125 µL/sample was immediately acquired on a CytoFLEX LX cytometer. To assess ADCC against SNU-C5 cancer cells from flow cytometry data, the relative decrease in viable cancer cells (EpCAM+, DAPI and Annexin V double-negative cells) was calculated based on unstimulated and antibody-treated samples as: ΔTumour cells_FACS_[%] = (%unstimulated − %treated)/%unstimulated × 100%.

### 2.7. Data Analysis and Statistics

Unless stated otherwise, all results in this study are reported as means (µ) with the corresponding standard error of the mean (SEM), calculated from technical and/or biological replicates. Statistical comparisons between two dependent or independent groups were performed using either a two-tailed Mann-Whitney U test or the Brunner-Munzel test. Comparisons between three groups were made by one-way ANOVA tests followed by a Tukey test to correct for multiple comparisons. For the exploratory analysis of 154 parameters of NK cell functionality, we omitted methods for multiple testing to reduce the risk for false-negative results (type II errors) while accepting a higher risk of false-positive results (type I errors) at a maintained 0.05 significance level. These assumptions are recognized for exploratory, hypothesis-generating studies since type II errors from adjustments after multiple testing significantly increase in studies where the number of analysed parameters exceeds the number of specimens [18]. Data analysis was performed using R (v.4.2.1) with RStudio (including the packages brunnermunzel, caret, dplyr, faraway, ggord, ggplot2, klaR, magick, magrittr, party, tibble, tidyr, tidyverse and Rtsne) or GraphPad Prism version 10.0.0 for Windows (GraphPad Software, Boston, MA, USA).

## 3. Results

### 3.1. Flow Cytometric Prediction of FcγRIIIa-V158F Phenotypes

Currently, the only reliable techniques to assess FcγRIIIa-V158F polymorphisms are by performing real-time PCR followed by melt curve analysis, TaqMan allelic discrimination assay or sequencing of the polymorphic exon 4 that requires specialized instruments, needs well-trained personnel and is rather time-consuming. Hence, in addition to performing the PCR-based approach to form our ground truth, we strived to develop a flow cytometry-based assay that is easy to use, has a short turn-around time of about 30 min and uses only a small amount of patient material (Figure 1A). In contrast to earlier approaches [19] we aimed to detect the FcγRIIIa-V158F polymorphism in a single staining. Therefore, we took advantage of the differing binding epitopes of two antibodies against the FcγRIIIa (CD16), namely MEM154 and LNK16 (Figure 1B) [3,19,20]. Anti-CD16 clone MEM154 binds the FcγRIIIa receptor at an epitope in close proximity to amino acid residue 158 and its binding is therefore heavily influenced by a potential amino acid exchange at this position. In the presence of a valine (V) residue, MEM154 binds very strongly, whereas its binding is almost completely abolished in the presence of a phenylalanine (F) residue. While clone LNK16 is still described to block Fc-FcγRIIIa interactions, its epitope is located in the C’ beta-sheet of the membrane-proximal domain further away from residue 158 and thereby can bind to FcγRIIIa’s bearing both amino acids. Yet, our analyses indicate a higher staining intensity when donors with homozygote F-residues were stained with the LNK16 clone yielding an even higher ratio between MEM154 and LNK16 binding depending on the FcyRIIIa-V158F polymorphism.

To test the ability to predict FcγRIIIa polymorphisms based on MFI information of both antibodies, we set MFI boundaries based on individuals classified as VF by PCR. When applying our acceptance criteria defined in Section 2, our analyses of samples from 91 individuals (39 healthy volunteers and 52 FIRE6 patients) revealed that the simple comparison of binding intensities of both antibodies was insufficient to reliably distinguish the three FcγRIIIa genotypes (Figure 1C): By applying optimal class boundaries based on the MFIs for MEM154 and LNK16 only 68% of all samples were classified correctly, with 11% of samples being misclassified between the low-affinity FF phenotype and the high-affinity VF/VV phenotypes.
Figure 1**Detection of FcγRIIIa-158 phenotypes by flow cytometry.** (**A**) Summary of the assay development to detect the FcγRIIIa-V158F polymorphism including sample source, preparation, and bioinformatics. FcγRIIIa-typing was established on 39 healthy donors followed by validation on a cohort of 52 mCRC patients from the FIRE-6 study. At each point, FcγRIIIa polymorphisms were detected by PCR sequencing and flow cytometry. (**B**) Gating strategy to detect FcγRIIIa-V158F phenotypes using two different CD16 clones. After selecting lymphocytes, NK cells were identified as CD3-CD14-CD56+ cells, and LNK16 and MEM154 binding was analyzed. Representative examples for each FcγRIIIa phenotype are shown. (**C**) Scatter plot of MEM154 and LNK16 binding (MFI) on NK cells. Dots represent individuals and FcgRIIIa-158 phenotypes are color-coded.
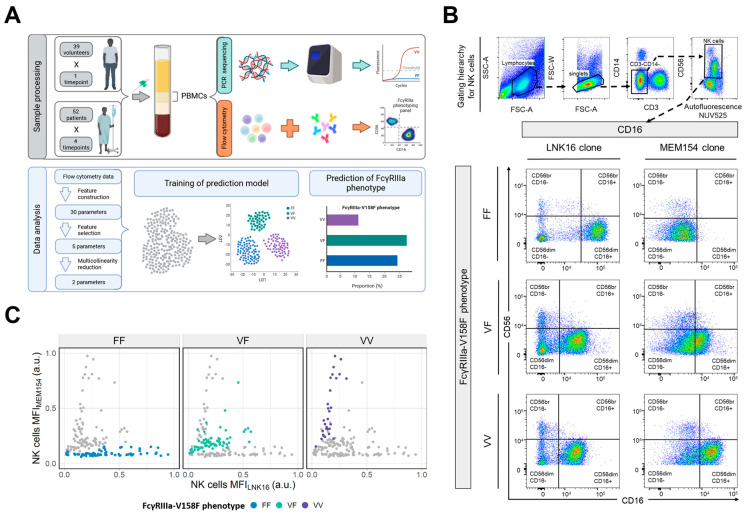


To advance our downstream analysis and increase the prediction precision we included additional parameters obtained from our flow cytometry panel. In total, we constructed 30 parameters attributed to the binding of the MEM154 and LNK16 clones on different NK cell subsets (CD56bright; CD56dimCD16+, CD56dimCD16−) including MFIs, frequencies and ratios thereof Appendix A). After identifying the ratio of the frequency of MEM154+ to LNK16+ CD56dim NK cells (rF_NKdimML_) and the ratio of the MFI of MEM154 to LNK16 signal on CD56dim NK cells (rMFI_NKdimML_) as the two most discriminative features using a conditional random forest model and subsequent reduction in multicollinearity by introducing a variance inflation factor cutoff, we built a supervised machine learning model by performing a Linear Discriminant Analysis (LDA) followed by Logistic Regression based on the data obtained from measuring PBMCs from 39 healthy donors. The mean prediction accuracy of the final model as assessed by 10-fold cross-validation was 94%, hence exceeding ≥ 90% prediction accuracy (acceptance criterion (1)). An example of distinct clustering of all FcγRIIIa phenotypes by the LDA model generated for a specific training set is shown in Figure 2A.

Next, we tested the developed model on PBMCs obtained from 52 mCRC patients treated within the Fire-6 study. While misclassification between the high-affinity VF and VV individuals did occur in some cases, 100% of FF individuals were classified correctly (fulfilling acceptance criterion (2)) (Figure 2B). During the FIRE-6 study, patients were treated with chemotherapy containing the IgG1 antibodies cetuximab and/or avelumab. Due to their potential to engage with FcγRIIIa, antibody treatment might interfere with the developed detection method. However, the mean prediction performance of the established flow cytometric assay as defined by weighted F1 scores after leave-one-out cross-validation was stable during therapy in the range from 91% to 95% (Figure 2C), indicating that FcγRIIIa characterization is possible at any time during treatment. Inter-cycle analyses again revealed that misclassifications mainly affected VV individuals while FF samples were correctly predicted at all timepoints. Since flow cytometric assays depend on the specific flow cytometry device used as well as its calibration and acquisition settings, we elucidated the minimal sample size needed to reconstruct our prediction model. To do so, we gradually reduced the sample size used to train the machine-learning model down to a minimum of 3 training samples. At every step, a new model was repeatedly fit for a random training set and evaluated on a fixed test set (repetitions *n* = 100) to minimize a possible bias due to picking “favorable” training samples by chance. As a rule for sample selection we forced the model to select samples at two different FcγRIIIa phenotype proportions: (1) FF:VF:VV of 4:4:2, mimicking the reported prevalence within the Caucasian population [21,22] and enabling random recruitment of donors to set up the model; (2) FF:VF:VV of 1:1:1, an equal distribution was thought to potentially reduce the sample size needed. As shown in Figure 2D, both approaches reached prediction accuracies of >90% when we used samples from at least six donors for model generation. For nine donors or more, the predictive performance already plateaued at the 94% level (dotted line in Figure 2D). These results indicate that measuring six to nine randomly chosen training samples might be sufficient to implement a reliable FcγRIIIa-V158F prediction model.

### 3.2. Impaired NK Cell Functionality in mCRC Patients

To set up a reliable assay to determine cytotoxic NK cell activity using scarce clinical samples we developed a 24 h co-culture experiment consisting of PBMCs and cells from colorectal cancer cell lines. We believe that investigating antibody-mediated immune reactions using whole PBMCs instead of purified NK cells is an advantageous approach as it better reflects the patient’s individual immune constitution during malignancy and allows for analysis of potential immunologic alterations and interactions during therapy. We assessed cytotoxicity by combining the rather easy but sensitive detection of LDH release in dying cancer cells with a sophisticated flow cytometry panel that additionally allows the quantification of several markers on NK, T, antigen-presenting and cancer cells (Figure 3A). During the FIRE-6 study, colorectal cancer patients were treated with the anti-EGFR antibody cetuximab alone or in combination with the anti-PDL1 antibody avelumab. Therefore, we screened a panel of 13 different human colorectal cancer cell lines in terms of signal:noise ratio in LDH release, surface expression of ADCC-relevant molecules and ADCC sensitivity (Appendix A). Additionally, we identified 3 × 10^4^ cancer cells as the lowest input and an effector to target (E:T) ratio of 10:1 to reliably detect ADCC (Appendix A). Throughout those analyses, SNU-C5 emerged as the most suitable target cell line in our model due to their ideal signal:noise ratio in LDH release over a large cell count range and a high expression of the antibody target molecules, EGFR and PDL1, resulting in detectable ADCC by cetuximab and avelumab (Appendix A). Their expression of co-stimulatory molecules like CD137L and CD40 and comparably low levels of ADCC counter-regulators MICA/B, HLA-E and HLA-A, might also explain the ADCC sensitivity of SNU-C5 (Appendix A).

To delineate antibody-mediated anti-cancer immune reactions by flow cytometry, samples were stained with the epithelial cancer cell marker EpCAM to identify cancer cells (Appendix A) together with immune lineage markers CD14, CD3, CD56 to differentiate effects on cancer cells, monocytes, T and NK cells, respectively (Figure 3B). We also included markers to detect regulations of checkpoint molecules on cancer cells and monocytes (PDL1, CD40) as well as checkpoint and activation markers on T and NK cells (CD107a, CD137, PD1, NKG2D, NKG2A, CD62L) (Figure 3C). ADCC strongly correlated between detection by LDH release and reduction in viable cancer cells as detected by flow cytometry confirming the suitability of the applied assay system (Figure 3D).

To display the usability and clinical significance of the developed protocol we assessed the capacity of antibody-mediated immunity in 35 patients with mCRC and compared them with 10 healthy donors that were matched for age (63y [47–81] vs. 64y [57–71]) and FcγRIIIa-V158F polymorphisms (FF/VF/VV = 42/40/17% vs. 38/38/23%). Comparison of the combined set of patient and healthy donor samples confirmed the well-described paradigm that donors bearing the FcγRIIIa-158FF polymorphism show a significantly reduced ADCC capacity while we did not detect significant differences between VF and VV donors (Figure 4A) nor between FcyRIIIa groups for any other NK cell activation marker. During the FIRE-6 clinical study, patients were initially treated with cetuximab followed by a maintenance therapy also containing avelumab. Since both antibodies are of the IgG1 isotype with the potential to engage FcγRIIIa, we aimed to show effector mechanisms of cetuximab and potential synergies between both antibodies in our assay set-up. As shown in Figure 4B, cetuximab single-treatment induced ADCC accompanied by CD16 downregulation and increased NK cell activation (CD137) and degranulation (CD107a). While the addition of avelumab seemed to have no direct effect on further increasing ADCC against SNU-C5, combinatory treatment enhanced the downregulation of CD16 and inductions of CD137 and CD107a indicating a potential synergistic NK cell activation by the treatment regime. In contrast to NK cells, T cells barely express FcγRIIIa and therefore should only be incidentally activated by cetuximab or avelumab. While our analysis revealed a slight increase in CD107a expression to a maximum of 2% CD107a+ T cells, the lack of altered CD137 or PD1 expression (Appendix A) confirmed NK cells as the main drivers of antibody-mediated cancer cell lysis.

Next, we aimed to compare the cetuximab-induced immune reactions between healthy donors with baseline samples from the FIRE-6 cohort. Using an explorative analysis approach of our in-depth flow cytometry panel, we analyzed changes in 77 individual parameters either as difference (delta, ∆) or fold change (FC) between unstimulated and antibody-treated samples resulting in a total of 154 measures. Unsupervised clustering using dimensionality reduction by tSNE revealed distinct clustering of samples from patients and healthy donors demonstrating strong differences in immune regulations in both populations (Figure 4C). Investigation of significantly differently regulated parameters (Appendix A) showed that samples from mCRC patients induced less effective ADCC as measured by a reduced LDH release (Figure 4D) or induction of dead (DAPI+AnnexinV+) EpCAM+ cancer cells (Figure 4E). Diminished NK cell activation was also reflected by a less prominent CD16 downregulation and less induction of CD137 (Figure 4D). In addition to NK cells, monocytes are another immune population expressing FcγRIIIa and whose activation is associated with PDL1 expression. Hence, the reduced induction of PDL1 on monocytes in mCRC patients (Figure 4E) indicates that besides NK cells other immune cells show an impaired cetuximab response. Finally, we also observed some changes pointing to cell interactions. While PBMCs from healthy donors eradicate PDL1+ cancer cells or induce downregulation of PDL1 on cancer cells (Figure 4E), levels of PDL1+ cancer cells remain unchanged in mCRC samples (Appendix A). Further, we detected higher frequencies of PD1+ NK cells in cancer patients (Appendix A). Despite we could not find direct correlations between ADCC and frequencies of PD1+ NK cells, the potential interactions between PD1 expressed on NK cells and PDL1+ cancer cells might be one explanation for the observed impaired NK cell functionality.

Together we could show that mCRC patients exhibit a reduced NK cell functionality, which highlights the potential of our established protocol to identify and track therapy-relevant immunological predispositions and regulations concerning therapies with anti-cancer mAbs.

## 4. Discussion

Immunotherapies involving mAbs, including targeted therapies and immune checkpoint inhibitors play a pivotal role in reinvigorating anti-cancer immune responses. While prior research has emphasized the clinical significance of FcγRIIIa-mediated immune activation through IgG1 isotype antibodies, comprehensive investigations of potential interactions between different components of the immune system and their interplay with cancer cells have not been fully explored [1,23]. In this study, we developed and validated a flow cytometric assay designed to identify FcγRIIIa-V158F polymorphisms and developed a further flow cytometric assay to investigate antibody-induced anti-cancer immune responses. Through a comparative analysis of patient samples from the FIRE-6-Avelumab phase-II clinical study and age-matched healthy donors, we identified NK cell dysfunctions in cancer patients.

The FcγRIIIa-V158F polymorphism was shown to have diverse clinical implications, including susceptibility to autoimmune disorders, infectious diseases [24,25], and predicting therapy responses to mAbs [26,27,28,29], cancer vaccinations [30] and oncolytic adenoviruses [31]. Hence, a broadly applicable assay to identify FcγRIIIa-V158F polymorphisms has a large area of application not only in clinical trials but also in routine practice. In contrast to an earlier two-sample protocol [19], we took advantage of two non-interfering antibody clones against CD16 either binding irrespective of FcγRIIIa-V158F polymorphisms (LNK16) or to a specific epitope only found in the valine-containing sequence (MEM154). Comparable to the approach by Böttcher and colleagues [19], we observed great variances of MFIs from both antibodies questioning the simplistic model based on MFI thresholds for two CD16 clones in clinical samples which is also one of the major challenges for broadly applicable flow cytometric analyses. To address these problems, we trained a machine-learning model to precisely predict the FcγRIIIa polymorphism based on flow cytometry data. In 10-fold cross-validation, we were able to correctly assign the FcγRIIIa polymorphism with a weighted F1 score of 94%. It is worth noting that all misclassifications exclusively occurred between the VF and VV groups. Since we and others [32,33,34] observed that a heterozygote valine at FcyRIIIa-158 is sufficient to unleash the full potential of NK cell-mediated ADCC, these misclassifications are unlikely to impact in vitro reactivity or clinical decision-making. Our study reports the first flow cytometry approach to identify FcγRIIIa-V158F polymorphisms based on a machine learning model and validated on clinically relevant samples, but broad application of flow cytometric detection also comes with several pitfalls. We intended to develop a rather simple staining protocol enabling its accordance with most flow cytometers. However, this technique still needs a high level of standardization such as sample preparation, quality controls, adjustments to laser configurations and centralized or well-trained personal for gating and data analysis [35]. Even if limited to one device, in an attempt to show the suitability of our approach to be trained in other laboratories, we were able to show that nine well-characterized measurements are sufficient to set-up our model. Furthermore, the recent emergence of automated gating strategies [36] might enable additional improvements by limiting operator variabilities; hence, increasing its application in scientific and clinical practice.

Induction of ADCC is an important mode of action of anti-cancer mAbs and has already been shown to have predictive value in colorectal cancer [37]. While we were able to verify the advantage of high target molecule expression for effective ADCC [38,39] and a provides correlation between EGFR and PDL1 expression [40], our comparison of 13 different CRC cell lines provide further insights into cetuximab-mediated ADCC. NK cells are able to recognize and kill HLA class I deficient cells like virus-infected or malignant cancer cells and are tightly regulated by co-stimulatory and inhibitory signals [41]. Hence, the observed inverse relationship between cetuximab-mediated ADCC and HLA-A expression reflects the nature of NK cell activation and aligns with a study in non-small cell lung cancer [42]. In contrast, the highest ADCC was detected against cancer lines expressing high levels of CD137L which in combination with the increased expression of CD137 on NK cells after cetuximab stimulation indicates the positive effect of co-stimulatory signals in enhancing ADCC. The potential of harnessing the CD137-CD137L axis to boost ADCC has been previously explored [43,44] and might hold promise to reactivate dysfunctional NK cells in cancer patients. NK cells from CRC patients exhibit reduced levels of activating NK cell receptors such as NKG2D, DNAM-1, NKp30, NKp44, NKp46 and cytotoxic perforin, and elevated levels of inhibitory receptors like NKG2A, CD85j and KIR3L1, indicating a diminished functionality [45,46,47]. Consequently, NK cells from oesophageal squamous cell carcinoma, breast cancer and colorectal cancer were already shown to exert reduced anti-cancer activity induced by trastuzumab or cetuximab [45,48,49]. Our study confirms impaired cetuximab-mediated NK cell functionality illustrated by reduced ADCC, CD16 downregulation and CD137 induction in mCRC patients. One possibility for the reduced ADCC is the elevated expression of PD1 on NK cells from cancer patients [50] which dampens their lytic activity and cytokine secretion [51]. Our findings show that NK cells from healthy donors, expressing significantly less PD1, exhibit effective lysis of PDL1-positive SNU-C5 cancer cells. In contrast, NK cells from mCRC patients, characterized by elevated PD1 levels, fail to eliminate PDL1+ SNU-C5, suggesting the involvement of the PD1-PDL1 axis in impairing cetuximab-induced ADCC. Additionally, previous studies indicate that cetuximab might foster immune interactions. For instance, cetuximab can skew monocytes from an immunosuppressive M2 to a more anti-tumorigenic M1 phenotype whose activation was accompanied by PDL1 upregulation [52]. Hence, our results of significantly reduced induction of PDL1 on monocytes from mCRC patients suggest that dysfunctions in immune cells that are relevant to promoting cetuximab effects extend beyond NK cells.

The presented results might offer several implications for future clinical trials. Differences in cetuximab reactivity in mCRC patients might serve as one explanation for the controversial results regarding the importance of ADCC in the clinic. It was shown that cancer progression [45] and different chemotherapies [53] have strong impacts on NK cell functionality and hence, might interfere with the simple assessment of FcγRIIIa-V158F polymorphisms to predict mAb-induced ADCC in vivo. The sophisticated assessment of phenotypic and functional parameters of a wide range of mAb-induced anti-cancer immunity might therefore offer a better picture of the clinical reality. Additionally, while IgG1 mAbs are known to enable strong ADCC in optimized in vitro settings, the presented impaired NK cell functionality in mCRC patients suggests that combinatory treatments might be needed to unleash the full potential of NK cell-focused therapies. Therefore, antibodies blocking inhibitory NK cell receptors might be advantageous to improve cetuximab-mediated ADCC [54] and the use of cytokines such as IL2 and IL15 has previously been shown to restore NK cell functionality in CRC patients [45,55].

## 5. Conclusions

In conclusion, we developed a novel assay set-up to identify clinically relevant phenotypic and functional parameters to define NK cell functionalities using two flow cytometry panels. We applied our approach to a cohort of mCRC patients and were able to detect immunological dysfunctions in cancer patients. Thus, our findings might help uncover immunological effector functions of anti-cancer mAbs, contribute to the development of combinatory immunotherapies and enable patient stratification.

## Figures and Tables

**Figure 2 cells-14-00032-f002:**
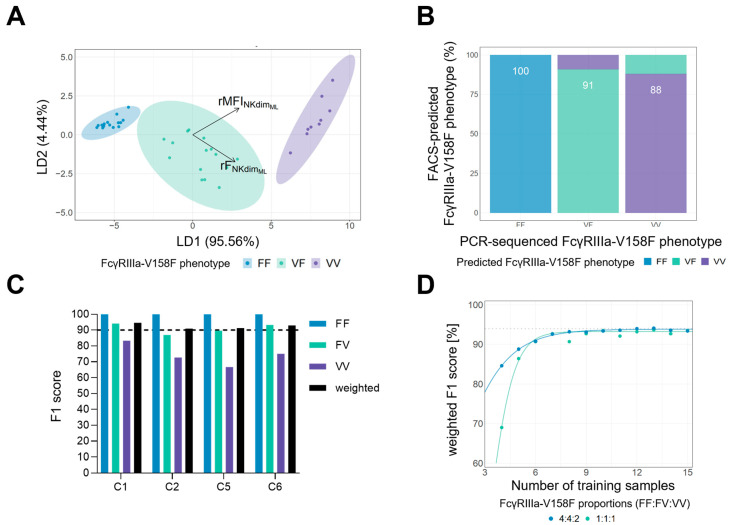
**Machine learning model predicts FcγRIIIa polymorphisms.** (**A**) Visualization of the generated LDA model for a specific training set to which Logistic Regression was subsequently applied for prediction. As highlighted, the major contributors for distinct clustering were the ratios between both CD16 clones (M = MEM154, L = LNK16) either in terms of frequency (F) or mean fluorescent intensity (MFI) on CD56dim NK cells. (**B**,**C**) Bar graphs show the mean prediction performance as defined by F1 scores of 10-fold cross-validation for the flow cytometry assay regarding all FcγRIIIa phenotypes (**B**) or the F1 scores of leave-one-out cross-validation per time point during therapy for all FcγRIIIa phenotypes and weighted for the study cohort (**C**). Highlighted are the correctly assigned FcγRIIIa phenotypes with respect to PCR-based detection. (**D**) Performance of prediction model built from gradually smaller size of training sets. For each sample size, a new prediction model was repeatedly trained and validated on the same testing set (repetitions *n* = 100) using two different approaches for FcγRIIIa phenotype proportions: approach 4:4:2 (blue) simulates the approximate prevalence of each phenotype in the Caucasian population, whereas approach 1:1:1 (green) assumes equal proportions of each phenotype in the training set.

**Figure 3 cells-14-00032-f003:**
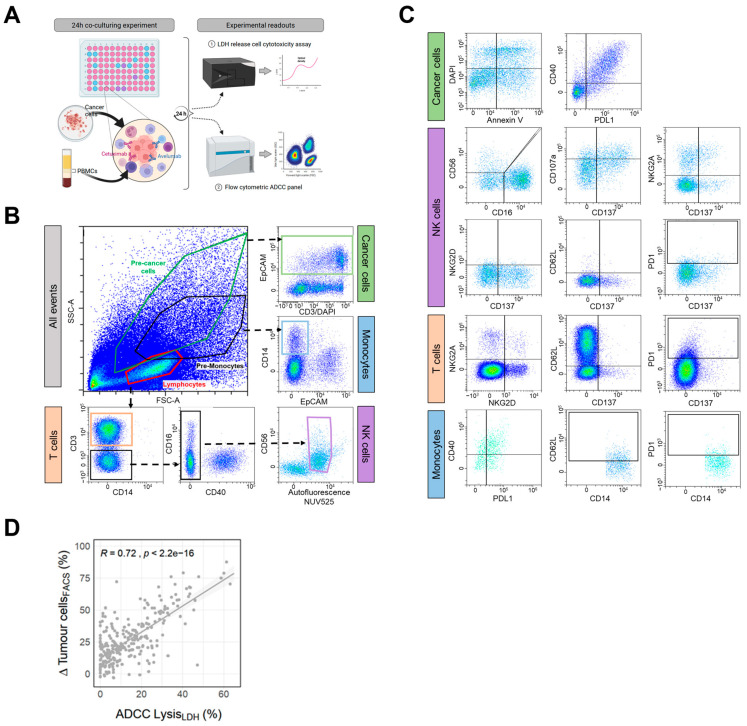
**ADCC assay development and gating strategy.** (**A**) Schematic representation of the established assay set-up. Briefly, 3 × 10^4^ SNU-C5 cancer cells were co-cultivated with 3 × 10^5^ PBMCs and stimulated with 100 ng/mL cetuximab or avelumab for 24 h. Anti-cancer response was then analyzed by LDH release and flow cytometry. (**B**,**C**) Gating strategy to analyze the viability of cancer cells and activation and regulations of immune checkpoints on immune cells. (**B**) After doublet exclusion (comparable to Figure 1) and physical discrimination using control samples with either cancer cell or PBMCs alone for pre-gating of respective cell types, EpCAM+ cancer cells, CD14+ monocytes, CD3+ T cells and CD3-CD14-CD56+ NK cells were distinguished. (**C**) For cancer cells, viability was detected by DAPI and Annexin V staining along with checkpoint expression of PDL1 and CD40. CD56+ NK cells were separated in CD56dimCD16+ and CD56hiCD16low/−. All NK cell subsets were analyzed for CD107a, CD137, NKG2A, NKG2D, CD62L and PD1 expression while T cells were measured for NKG2A, NKG2D, CD137, CD62L and PD1 expression. CD14+ monocytes were gated for CD40, PDL1, CD62L and PD1 expression. (**D**) Scatter plot of ADCC values from experiments with PBMCs from healthy donors or mCRC patients detected by LDH release assay (ADCC Lysis_LDH_) or flow cytometry (∆Tumour cells_FACS_). Each dot represents matched values from both readouts and the Pearson’s correlation coefficient together with the *p*-value for correlation fit is depicted.

**Figure 4 cells-14-00032-f004:**
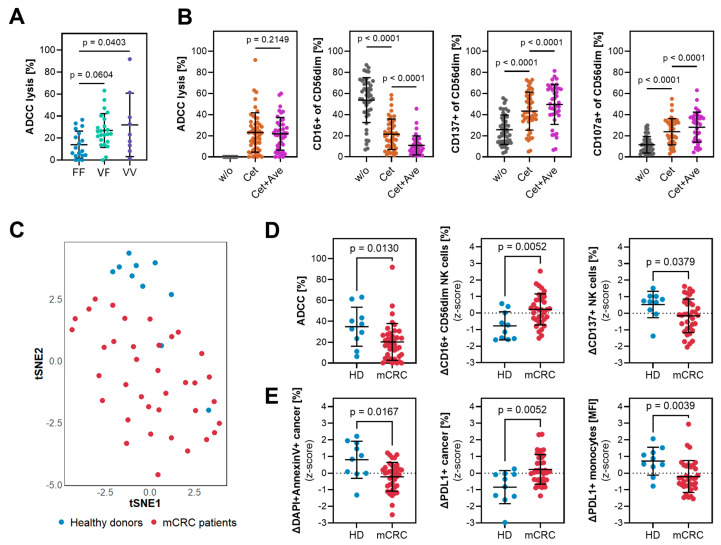
**Reduced cytotoxic potential in mCRC patients.** (**A**,**B**) Healthy donors (*n* = 10) and baseline samples from mCRC patients (*n* = 35) of the FIRE-6 study prior to therapy initiation were grouped according to their FcγRIIIa polymorphism and (**A**) assayed for cetuximab (Cet) mediated ADCC against SNU-C5 cancer cells. Additionally, samples from these donors were also stimulated with a combination of cetuximab and avelumab (Cet+Ave) to assess (**B**) ADCC and regulations of CD16, CD137 and CD107a expressions according to different stimulations. (**C**) After z-score standardization, t-Distributed Stochastic Neighbor Embedding (tSNE) dimensionality reduction of 154 flow cytometry parameters shows the response to ex vivo cetuximab stimulation in healthy donors (blue, HD) or mCRC patients (red, mCRC). (**D**,**E**) Examples of differentially regulated parameters representing (**D**) NK cell functionalities or (**E**) regulations on cancer cells and monocytes. ∆-values are calculated as the difference between unstimulated and cetuximab-treated samples. Statistics: (**A**,**B**) One-way ANOVA followed by Tukey’s multiple comparison test. (**D**,**E**) Mann–Whitney U test. (**A**,**E**) Each dot represents the mean of technical triplicates from individual donors.

## Data Availability

Data, scripts and material not included in the manuscript will be made available to qualified researchers on reasonable request to the corresponding authors.

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
