# Peer review of "Flow Cytometric Assessment of FcγRIIIa-V158F Polymorphisms and NK Cell Mediated ADCC Revealed Reduced NK Cell Functionality in Colorectal Cancer Patients"

_cells, 2024, doi:10.3390/cells14010032_

Round 1

Reviewer 1 Report

Comments and Suggestions for Authors

In this paper, Schiele and co-workers describe two complementary assays using flow cytometry. One assay applies a machine learning tool to multicolor flow cytometry measurements to predict the G559T genotype of the FCG3A gene at the level of peripheral blood cells, while the other approach uses flow cytometry to assess NK-cell function and immune-tumor cell interactions during cancer therapy.

I found several weaknesses that I listed below:

Major Points

Q1. I have noticed that throughout the text the term 'polymorphism' is used indiscriminately as a synonym for genotype or aminoacidic substitution. The G559T polymorphism in the FCGR3A gene results in the aminoacidic substitution of valine by phenylalanine at position 158 of the protein. As a result, the title and text need to be changed accordingly.

Q2. I found the abstract unclear, some information is taken for granted and there are too many unspoken acronyms that make the reading not very fluent.

Q3. Lines 55-58: Please, add the reference.

Q4. Line 156: Please, indicate the clone names and refer to the corresponding Table.

Q5. Lines 160, 176 and 276: The authors declare that they analyzed 81 samples (L160) or 82 samples (in lines 176 and 276), from 39 healthy individuals and 52 patients. The sum of individuals is 91. Please, clarify or correct.

Q6. Line 243. Statistical analysis: what is the level of significance considered? Have you used any corrections according to the multiple tests used (e.g. Bonferroni correction?).

Q7. Table S4, I suggest changing the names of the traits to the original names as used in the figures, because the names used complicate the interpretation of the data. For example, the name “MFI of M on M+ NK dim” could be: “MEM154 on MEM154+ CD56dim Nk cells”.

Q8. Line 346: Are PBMCs from patients? Please clarify.

Q9. Line 369: Figure 2 is actually Figure 3, and Figure 3 is actually Figure 4.

Q10. Line374. In Figure 2b (actually figure 3b), in the scatter plot analyzing the morphological parameters (FSC vs. SSC), it is not very clear how the three populations were discriminated. There are so many events and the gates partially overlap. Furthermore, what is meant by pTC? Please substitute the following names: lympho and mono as lymphocytes and monocytes, respectively.

Q11. Lines 455-459. Please, add the reference.

Q12. Regarding the effect of treatment with cetuximab and/or avelumab on ADCC induction and NK cell activation, did you find differences between patients stratified by FCGR3A genotype?

Q13. I suggest adding a table to summarize the results obtained from functional assays indicating conditions tested, p-values etc.

Minor points:

Q14. Throughout the text there are several acronyms that have not been made explicit, others are not made explicit the first time they are used but later in the text. Please change them consistently throughout the text.

Q15. Figure 1 B. Please, try not to cover the figure with the names of the cell populations.

Q16.In tables S1, S2 and S3 it would be appropriate to indicate the amount of antibody used rather than the dilution.

Reviewer 2 Report

Comments and Suggestions for Authors

The manuscript submitted by Schiele and coworkers describes an innovative method to detect FcγRIIIa-V158F polymorphism and to assess NK cell-mediated ADCC, both by flow cytometry. The approach is overall based on simple techniques which allow its applicability in several pathological context in which ADCC assessment can be tested to predict clinical outcome. Moreover, Schiele and coworkers propose some mechanisms possibly involved in ADCC impariment observed in mCRC patients, which could be exploited to develop secondary/combined therapies to improve the outcome of these patients. Moreover, the methods are described in great detail, which guarantee the reproducibility. Overall, I believe that this work is of high interest in itself and displays a good level of translationability in different pathological contexts as well as in clinical practice. Nevertheless, there are two major points that need to be revised.

At methodological level, the evaluated the differences in 154 parameters between healthy donors and patients one by one by using Mann-Whitney or Brunner-Munzel test without any further correction ( I refer to data shown in Figures 4D, 4E and S3B). This is not entirely correct, since multiple tests can lead to spurious significances. Therefore, Bonferroni’s correction should be applied (in this case 0.05/154 = 3e10-4) and only the parameters whose p-value is below the correction should be kept as significant (which means that the parameters shown in Figure 4D and 4E would be not significant any more). It is well recognized that Bonferroni's correction can be too stringent in some instances, but the authors should still  provide an explanation regard they kept p-value significance at 0.05. For this point, the authors can refer to 10.3389/fimmu.2022.811131.

Since a fundamental part of the work is devoted on developing a method for FcγRIIIa-V158F assessment, the  discussion should be improved by a more in depth comparison between the proposed strategy and the method previously developed by Bottcher et al as well as with PCR-based strategies to detect FcγRIIIa-V158F polymorphism. Indeed, these points are shortly addressed in Paragraph 3.1, and only advantages are mentioned. However, flow cytometry is a complex technique as well, with many aspects that need to be standrdized and/or correclty managed. For example, laser voltages for optimal stain index usually vary between instruments, which prevents the possibility to exploit the same settings. Moreover, gating design and compensation are often operator-dependent, which highlights the need of trained and skilled personnel. Therefore, together with the advantages of this flow cytometry approach comapred to previous strategy and PCR, the authors should also highlight also the limitations of their method as well as the aspects that might need of standardization/optimization, especially for those readers who are not familiar with flow cytometry.    

In lines 491-493, the authors say that HLA-A expression on cancer cells inversely correlates with their sensitivity to ADCC. Honestly, based on data reported in Figure 2SB, such a correlation is not so apparent. Indeed, the major part of the tested cell lines seem to have similar expression (10/13 cell lines go from white to light blue) and frequency (9/13 cell lines are pink) of HLA-A. Accordingly, the two parameters are on the right of the Figure, which shows the less correlated parameters. Therefore, the sentence should be amended according the obtained evidences.

Minor points:

Table S1 includes features of the FcγRIIIa panel, not the clinical features of patients, which should be included as indicated in the text

In Figure 1B the authors identify NK cells by using CD56 vs. NUV525, while in Figure 2B NK cells are identified by CD56 vs. autofluorescence. Is it the same strategy? In case, I recommend to use the same nomenclature, possibly combining the two ways as autofluorescence (NUV525)

Panel C in Figure S3 is indicated as B. Please correct

Figure 3 in incorrectly mentioned in the caption as Figure 2. Similarly, the caption of Figure 4 says Figure 3 and should be amended

In Figure 4B, the p-value between w/o and cet for ADCC lysis is missing and should be added

Reviewer 3 Report

Comments and Suggestions for Authors

Summary

In this manuscript, the authors reported the development of a machine learning model to predict the V158F polymorphism of CD16 (FcgRIIIa) receptor based on flow cytometry measurements and established a 15-color flow cytometry panel to assess NK-mediated ADCC function when co-cultured with tumor cells in vitro. Their examination of ADCC activities of peripheral blood mononuclear cells (PBMCs) derived from healthy donors compared with metastatic colorectal cancer (mCRC) patients facilitated identification of correlations between expression of cell surface markers and ADCC efficacy. Their data demonstrated that impairment of ADCC mediated by cetuximab and/or avelumab in mCRC patients was associated with PD1 up-regulation in NK cells, indicating potential contribution by the PD1-PD-L1 signalling axis. This study aims to develop an alternative flow cytometry-based assay that is easy to implement and far less time consuming than current techniques to discriminate between FcgRIIIa-V158F polymorphisms based on real-time PCR followed by melt curve analysis or sequencing. The authors managed to correctly assign FcgRIIIa-V158F polymorphisms with mean prediction accuracy of 94% with misclassifications found to occur exclusively between VF and VV group and thus unlikely to impact in vitro cytotoxicity or clinical stratification of patients for cetuximab and/or avelumab treatment since heterozygote valine at FcgRIIIa-158 is sufficient to confer optimal NK-mediated ADCC. Notwithstanding the important progress which the authors made to increase the efficiency in precisely distinguishing FcgRIIIa-V158F polymorphisms, this reviewer lists below concerns which have to be addressed before this manuscript can be considered further for publication.

Major comments

1.       Figure 1B: Authors are requested to include singlet and viability gates in their flow cytometric gating strategy for NK cells. Without these gates, significant debris is apparent in the representative SSC-A vs FSC-A plot which will adversely impact the reliability of downstream data analysis and interpretation.

2.       Line 118: Authors stated that “Clinical characteristics of mCRC patients and healthy donors are summarized in Table S1.” However, Table S1 as provided shows “Flow cytometry antibodies of the FcgRIIIa panel”. Please provide the clinical characteristics of the healthy donors and mCRC patients used in this study.

3.       Figure S1: This reviewer appreciates the authors’ inclusion of antibody titration data. However, the clone of CD16 APC-Cy7 antibody shown is unclear as neither MEM154 nor LNK16 are conjugated to APC-Cy7 according to Table S1. Both MEM154 and LNK16 clones should be included for titration. Did the authors perform validation of selected markers such as CD56, NKG2D, CD107a, etc. which are highly expressed in NK92 cell line?

4.       Figure 1C: Authors are urged to explain more clearly how misclassification of 11% of PBMC samples was deduced from the scatter plots showing MFI of CD16 expression observed in the three V158F genotypes? Was there corroboration performed with sequencing of samples?

5.       Figure 2: Authors are requested to additionally provide more comprehensive metrics beyond mere accuracy to evaluate machine learning model performance, especially if class imbalances exist within the dataset. These can include at least one of the following: precision/recall scores, F1 scores, or a confusion matrix.

6.       Second Figure 2B which this reviewer is wrongly labelled and should be Figure 3B: Authors are again requested to incorporate singlet and viability gates in their flow cytometric gating strategy.

7.       Second Figure 2D which should be Figure 3D: Authors are requested to clarify if percentage change in tumour cells visualized via FACS was determined by tumour cells as a proportion of, e.g. total singlets, live cells, or events within the pTC gate.

8.       Figure 3A which should be Figure 4A: The data indicated that FcgRIIIa-V158F polymorphisms differentially affect ADCC. Furthermore, some of the data shown in Figure 3B-E contained obvious sub-clusters, e.g. % CD137+ of CD56dim in Figure 3B. Authors are urged to examine how VV/VF/FF polymorphisms differentially affect the cytotoxic potential and/or surface markers such as CD16, CD137, CD107a expression of NK cells co-cultured with tumor cells when treated or not with cetuximab and/or avelumab as depicted in Figure 3B-E. It will be instructive to ascertain if healthy compared with mCRC patients preferentially harboured  a reduced NK cell functionality

9.       Lines 347-350: This reviewer understands the authors’ rationale of using whole PBMCs instead of purified NK cells to assess cytotoxicity mediated by cetuximab and/or avelumab in order to better reflect the patient’s immune constitution during malignancy and analyze immunological interactions while undergoing therapy. However, pro-inflammatory CD16+ monocytes in patient’s PBMCs can also mediate ADCC and therefore confound interpretation of FcgRIIIa-V158F polymorphisms affecting ADCC function of only NK cells within PBMCs.

10.   Figure 3 which should be Figure 4: Are there any differences in the MFI and/or proportion of each NK or T cell marker/subset when comparing FF vs VF vs VV genotypes, and healthy donors vs mCRC patients?

11.   Figure 3 which should be Figure 4: For clarity, legend should include the mCRC cell line used for ADCC evaluation which is presumably SNU-C5.

12.   With the large number of variables (154) being evaluated between unstimulated and antibody-treated samples, the authors are strongly encouraged to perform multiple hypothesis correction, e.g. Bonferroni correction or Benjamini-Hochberg false discovery rate, to avoid falsely identifying variables as statistically significant due to random chance. Furthermore, because the fold change and the difference (D) in % of each cell subset are correlated measures, the authors could further reduce the number of variables/hypotheses tested by choosing either fold change or D for each parameter in their analyses for Figure 4C and Figure S3.

13.   Figure 4E: The authors’ point that “levels of PDL1+ cancer cells remain unchanged in mCRC samples” should be substantiated by showing the baseline expression levels of PDL1 on the cancer cells prior to co-culture with PBMCs.

14.   Lines 447 to 448: The authors postulated that “interactions between PD1 expressed on NK cells and PDL1+ cancer cells could result in the observed impaired NK cell functionality”. Is there a correlation between the proportion of PD1+ NK cells with the extent of ADCC observed, and does the extent of this correlation vary between FF vs VF vs VV or healthy donors vs mCRC patients?

15.   Additionally, this could be further investigated by performing the same co-culture experiment with and without the addition of antibodies blocking PD1/PDL1 binding.

16.   Lines 515 to 516: The authors observed a “reduced induction of PDL1 on monocytes from mCRC patients” and stated that this could be indicative of skewing monocytes towards “a more anti-tumorigenic M1 phenotype whose activation was accompanied by PDL1 upregulation”. They could further support this claim by reporting the expression of other M1-like markers on monocytes, and assessing whether increases in expression of these markers are more pronounced in healthy donors vs mCRC patients.

Minor comments

17.   Please correct the following typographical errors:

o  Page 5, line 247: “Mann-Whitney t-test” should be “Mann-Whitney U test”.

o  Page 11, line 426 (Figure 3 legend): “Mann-Whitney test” should be “Mann-Whitney U test”.

o  The manuscript text references Figure 4 (line 400 onwards), but no Figure 4 is provided in the manuscript. However, there are 2 figures labelled Figure 2. From the description in the text, Figure 4 seems to refer to the panels in Figure 3, and Figure 3 seems to refer to the panels in the second Figure 2 instead. Please verify and revise the figure numbers as appropriate.

o  Figure S3: There are 2 panels labelled “B”, one of them should be labelled “C” instead.

o  Page 21, line 778-779 (Figure S3 legend): “Mann-Whitney test” should be “Mann-Whitney U test”.

Round 2

Reviewer 1 Report

Comments and Suggestions for Authors

The authors have adequately answered my questions and I believe the manuscript can be published in this form.

The only note I would like to make concerns the answer to comment no. 16:"In tables S1, S2 and S3 it would be appropriate to indicate the amount of antibody used rather than the dilution."

I am sorry I did not explain myself well, but I meant the amount of antibody expressed in ug or ng. I agree with the authors that ul are not useful. I have no other comments.

Reviewer 2 Report

Comments and Suggestions for Authors

The authors addressed all my concerns. I really appreciated their work and their replies to my comment. 

Reviewer 3 Report

Comments and Suggestions for Authors

This reviewer is satisfied with authors' revisions which adequately address the concerns raised and recommend their revised manuscript for publication.